# Cyberbullying, Social Media Addiction and Associations with Depression, Anxiety, and Stress among Medical Students in Malaysia

**DOI:** 10.3390/ijerph20043136

**Published:** 2023-02-10

**Authors:** Michelle Hui Lim Lee, Manveen Kaur, Vinorra Shaker, Anne Yee, Rohana Sham, Ching Sin Siau

**Affiliations:** 1Hospital Sentosa, Kuching 93250, Malaysia; 2Faculty of Medicine, Universiti Malaya, Kuala Lumpur 50603, Malaysia; 3School of Psychology, Asia Pacific University of Technology and Innovation, Kuala Lumpur 57000, Malaysia; 4School of Business, Asia Pacific University of Technology and Innovation, Kuala Lumpur 57000, Malaysia; 5Centre for Community Health Studies, Faculty of Health Sciences, Universiti Kebangsaan Malaysia, Kuala Lumpur 50300, Malaysia

**Keywords:** cyberbullying, social media addiction, medical students, depression, anxiety

## Abstract

This study aims to determine the prevalence and factors associated with cyberbullying and social media addiction. This cross-sectional study was conducted among 270 medical students from a public university in Kuching, Malaysia. The instruments used in this study included the cyberbullying questionnaire survey, Bergen Social Media Addiction Scale (BSMAS), and the Depression Anxiety Stress Scale 21-item (DASS-21). The prevalence of cyberbullying victimization was 24.4%, whilst 13.0% reported cyberbullying perpetration over the past six months. Male gender was positively associated with both cyberbullying perpetration and cybervictimization, whilst social media addiction was positively associated with cybervictimization. Psychological motives such as positive attitudes toward cyberbullying and gaining power were associated with cyberbullying perpetration. Cybervictimization doubled the tendency to depression (a*OR* 2.50, 95% *CI* [1.23, 5.08], *p* = 0.012), anxiety (a*OR* 2.38, 95% *CI* [1.29, 4.40], *p* = 0.006), and stress (a*OR* 2.85, 95% *CI* [1.41, 5.77], *p* = 0.004), whilst social media addiction was associated with a higher tendency to depression (a*OR* 1.18, 95% *CI* [1.10, 1.26], *p* < 0.001), anxiety (a*OR* 1.15, 95% *CI* [1.08, 1.22], *p* < 0.001), and stress (a*OR* 1.21, 95% *CI* [1.12, 1.32], *p* < 0.001). Medical schools in Malaysia need policies and guidelines against cyberbullying.

## 1. Introduction

The internet and its associated technologies are becoming more widely used worldwide. There are 4.66 billion active internet users globally, constituting 60% of the global population, with the largest number of online users originating from Asia [1]. According to the Malaysian Communications and Multimedia Commission Internet Users Survey [2], there were 87.4% of internet users in the country in 2018, an increase in internet users from 76.9% in 2016. The same survey found that 24.6 million Malaysians were social networking users, and the most popular social networking sites in Malaysia were Facebook followed by Instagram and YouTube [2]. The proliferation of technology use has given rise to problematic behaviors associated with internet usage, such as social media addiction and cyberbullying [3,4].

Cyberbullying has been defined as bullying which is perpetrated through the medium of technology usage in online settings, such as on social media or through a text message [5,6]. Cyberbullying has also been characterized as an intentional, repeated, and aggressive act of inflicting harm on another person through electronic means of contacting the victim, and occurs over time [7,8]. Most of the research about cyberbullying has been linked to traditional bullying and is mainly studied in school children [9]. There is growing evidence that cyberbullying is becoming more common among children and adolescents. A systematic review by Zhu et al. [10] showed that cyberbullying perpetration among children and adolescents was between 6.0 and 46.3%, and cyberbullying victimization was between 13.99 and 57.5%. Cyberbullying is not isolated to schoolchildren and adolescents but also occurs among university students and young adults with the prevalence of cyberbullying ranging from 3 to 40% for cyberbullying perpetration and 7 to 62% for cyberbullying victimization [11]. This matter is complicated by the fact that a substantial number of cyberbullies were also cybervictims. A meta-analysis by Lozano-Blasco et al. [12] reported that there is a moderate-high correlation (*r* = 0.428) between being both a cybervictim and a cyberbully. Cyberbullying in university students may lead to long-term psychological distress, depression, lower self-esteem, and poor academic performance [13,14].

Previous studies on cyberbullying among Malaysian young adults reported that cybervictimization ranged from 18.6 to 66.0% and cyberbullying perpetration ranged from 8 to 35% [15,16,17]. Cyberbullying has been linked to the amount of time spent on the internet, with greater amounts of time spent online being associated with greater cyberbullying activities [18].

Social media addiction has been described as a type of behavioral addiction, similar to other problematic usages of internet applications such as smartphone addiction and internet gaming disorder [19]. A study found social media to be the main tool used by young adults for cyberbullying in Malaysia [15]. Problematic use of social networking sites has been associated with psychological distress, depression, anxiety, and low self-esteem [20,21,22].

Problematic social media use, including social media addiction, may pose a risk for increased cyberbullying and cybervictimization. Craig et al. [23] applied the problem behavior theory and social learning theory to elucidate this association. According to the problem behavior theory, risk behaviors usually occur together, and certain individuals have specific profiles which make them more vulnerable to risky behaviors. For example, individuals with social media addiction may have increased exposure to cyberbullies or cybervictims due to spending more time on these sites; thus, the addiction to social media increases the likelihood of cyberbullying to occur. Based on the social learning theory, individuals who are on social media platforms for a longer time may witness more aggression, and through the effect of modeling and reinforcement, may perceive aggressors to be of higher social status, and emulate these behaviors. The sense of belongingness of conforming to group norms may also reinforce group bullying behaviors. In a study across 42 countries among adolescents aged 11 to 15 years old, problematic social media use had modest to strong associations with cyberbullying and cybervictimization [23]. Structural equation modelling in a study among Malaysian university students showed that social media use was significantly associated with cyber engagement, and this engagement was also associated significantly with cyber harassment, cyberstalking, and cyberbullying [24].

Both cyberbullying and social media addiction have been associated with psychological distress [25]. A study among Bosnian adolescents in a state hospital showed that those diagnosed with anxiety and depressive disorders had higher scores for cybervictimization [26]. Another study among 1691 Malaysian adolescents revealed that both before and during the COVID-19 pandemic, those with depression symptoms had a higher tendency to experience cyberbullying [27]. A study among college students in China showed that cyberbullying in social media and gaming contexts was associated with higher anxiety and internet addiction symptoms [28]. Among university students in Malaysia, social media use during the COVID-19 pandemic has been linked to depression and anxiety symptoms [29] and lower self-esteem [22].

There is a paucity of studies addressing both cyberbullying and social media addiction among medical students. This is important as a meta-analysis showed that internet addiction, of which social media addiction is a subset [30], was detected among 30.1% of medical students [31]. Bullying in the medical education setting has been prevalent. In a review of 68 articles, 38.2% received undue pressure to produce work, and 36.1% were directed to work below their competency level [32]. Nearly half (46.2%) of trainee doctors in a UK study reported having been cyberbullied at least once [33]. Mental health among medical students has also been low. A meta-analysis found that the pooled prevalence of anxiety among medical students was 33.8% [34], whilst the pooled prevalence of depression was 28% in another meta-analysis [35]. Suicidality is a significant issue affecting medical professionals worldwide, indicating the need for early intervention [36]. A study among Malaysian undergraduate medical students found that 23.8%, 51.6%, and 15.9% reported depression, anxiety, and stress symptoms at the beginning of the semester [37]. Therefore, we aimed to determine the prevalence of cyberbullying perpetration, cyberbullying victimization, and social media addiction among medical students and their associated factors. Furthermore, this study aimed to examine the association between cyberbullying, social media addiction, and depression, anxiety, and stress among medical students.

## 2. Materials and Methods

### 2.1. Study Setting, Participants and Procedures

A cross-sectional study was conducted among Year 1 to Year 5 medical students enrolled in the Doctor of Medicine program in Universiti Malaysia Sarawak from September 2020 to November 2020. The participants were recruited via an online survey questionnaire using Google Forms. All medical students currently studying in Universiti Malaysia Sarawak were invited to participate via related online course platforms. Participants provided informed consent by ticking a box before they were allowed to proceed to answer the questionnaire.

Measures were taken to ensure the confidentiality of all personal data collected in the study, including the information disclosed in the questionnaires conducted. Participants with psychological distress who were identified during the study were referred to the appropriate services for further management.

### 2.2. Measures

The participants were given self-administered questionnaires to complete in English including a sociodemographic questionnaire, cyberbullying questionnaire survey, Bergen Social Media Addiction Scale (BSMAS), and Depression Anxiety Stress Scale 21-item (DASS-21). A sociodemographic questionnaire was developed to obtain relevant sociodemographic information on age, gender, race, religion, and year of study.

The cyberbullying questionnaire survey was developed in English by Balakrishnan in 2017 to research cyberbullying among young adults in Malaysia and consists of three parts [38]. The first part assessed internet usage and frequency, while the second and third parts focused on cyberbullying and the motives behind cyberbullying perpetration among Malaysian young adults [38]. The second part consisted of eight statements on cyberbullying incidences and the medium where cyberbullying took place whether via social media, chats, email, mobile phone calls, or messaging services. Each of the questions was measured based on the frequency of the incidence in four categories from never, rarely, occasionally to frequently. Participants who answered yes to either rarely, occasionally, or frequently to the question “I was cyberbullied before” were considered a cybervictim (yes). This also applied to the categories of cyberbullies and bystanders in order to have a general overview of the prevalence of cyberbullying irrespective of the frequency of being victimized, bullying others, or being a bystander. The third part consisted of 37 statements related to motives for cyberbullying which consists of three main domains of sociocultural (e.g., “Cyberbullies tend to cyberbully as their friends are doing it”), psychology (e.g., “Cyberbullying empowers one (by hurting others)”), and technology (e.g., “Messages or act of cyberbullying can be broadcast to a large audience”) [38]. The scale score of the sociocultural, psychology, and technology domains had internal consistency reliability of α = 0.872, 0.874, and 0.699, respectively.

The Bergen Social Media Addiction Scale (BSMAS) was used to assess the severity of social media addiction on the internet [39]. The BSMAS is a 6-item self-report Likert type scale that is brief, with good psychometric properties based on the dimensions of addictive behavior (salience, tolerance, withdrawal, mood modification, conflict, and relapse). Each item can be scored from 1 (*very rarely*) to 5 (*very often*) with a total score ranging from 6 to 30. Higher BSMAS scores imply a greater severity of social media addiction. A study in Hungary used a cut-off score of 19 and higher to determine the presence of social media addiction symptoms [40]. The BSMAS has been shown to have good psychometric properties and has been translated into different languages and validated in those languages [40,41]. The scale score had an internal consistency reliability of α = 0.837.

Psychological distress was screened using the Depression Anxiety Stress Scale-21 (DASS-21) which is a widely used screening tool to assess depression, anxiety, and stress in the community setting and has been demonstrated to have good validity and reliability among Malaysian university students [42]. The DASS-21 consists of the depression, anxiety, and stress subscales with 7 items per scale with scores ranging from 0 (*Did not apply to me at all*) to 3 (*Applied to me very much, or most of the time*). Each scale has been given recommended cut-off scores to assess the severity levels from normal, mild, moderate, severe to extremely severe [43]. The internal consistency reliability of the scale score was α = 0.955.

### 2.3. Statistical Analysis

The data were analyzed using the Statistical Package for the Social Sciences (SPSS), version 26.0 [44]. Descriptive statistics were used for the sociodemographic data, cyberbullying and tools used, motives of cyberbullying, social media addiction, and DASS-21 scores. Those who experienced either mild, moderate, severe, or extremely severe symptoms of depression were grouped together in another category as those who had depression. This rule was similarly applied to participants who scored mild to extremely severe on anxiety and stress scales and were recategorized as experiencing anxiety or stress.

Univariate analysis was first performed using simple logistic regression to find the association between the sociodemographic factors, motives behind cyberbullying, and social media addiction with cyberbullying perpetration and victimization. Simple logistic regression was also conducted to assess the association between cyberbullying perpetration, victimization, social media addiction, and their sociodemographic factors with depression, anxiety, and stress.

Multivariate analysis was done by including all independent variables with *p* < 0.25 from the univariate analyses into the multiple logistic regression model. Multiple logistic regression analysis was done to assess the association between cyberbullying perpetration, victimization, and statistically significant sociodemographic factors with depression, anxiety, and stress. All tests are two-tailed, with a significance level of *p* < 0.05.

### 2.4. Institutional Review Board (IRB) Approval Statement

Ethical approval was obtained from the Universiti Malaya Research Ethics Committee (UMREC; No.: UM.TNC2/UMREC-970) and the Medical Ethics Committee of the Faculty of Medicine and Health Sciences, Universiti Malaysia Sarawak (No.: FME/20/01).

## 3. Results

A total of 270 participants responded to this survey (mean age = 21.98 ± 1.54). The participants ranged from Year 1 to Year 5 with the most participants from Year 2 at 25.9% and the least participants from Year 1 at 11.1%. The majority of participants were female at 75.6%, and more than half were Muslims (56.3%). Half of the participants were Malay (49.6%), and Bumiputera Sarawak made up almost a quarter of the participants (23.7%). Most participants used the internet (66.3%) and used their mobile phones (64.1%) for more than five hours a day. More than half of the participants used social media for 1 to 5 h each day (59.6%), whilst 35.6% used it for more than 5 h each day.

The prevalence of being a cybervictim was 24.4% over the past 6 months. Thirteen percent reported having cyberbullied another person. Interestingly, more than two-thirds of the respondents (81.5%) had witnessed cyberbullying incidences. The majority of cyberbullying occurred over social media (89.3%), followed by messaging services (71.1%). The mean score for the BSMAS was 17.05 ± 5.05. Using the Hungarian study cut-off score of 19, [17] 102 participants, or 37.8%, had social media addiction symptoms. The prevalence of at least mild depression, anxiety, and stress symptoms were 34.0%, 38.5%, and 21.1% respectively (see Table 1).

With regards to factors associated with being a cyberbullying victim, the results of the multiple logistic regression showed that male medical students have 2.66 times the tendency to being a cybervictim (95% *CI* [1.34, 5.11], *p* = 0.003) compared to female medical students. Medical students who have higher levels of social media addiction with an increase in the total score of BSMAS by one have 1.09 times the tendency to be a cybervictim (95% *CI* [1.02, 1.16], *p* = 0.013). The cyberbullying motives of technology and psychology were initially significant from univariate analysis, but after adjusting for other factors were not significant (see Table 2).

Multiple logistic regression analysis showed that the male gender has 4.48 times the tendency to being a cyberbullying perpetrator (95% *CI* [1.94, 10.33], *p* < 0.001) and every increase in score by one on psychological motives increased the tendency to cyberbullying by 2.16 times (95% *CI* [1.26, 3.73], *p* = 0.005) (see Table 3).

Multiple logistic regression analyses testing the associations between cyberbullying and social media addiction with depression, anxiety, and stress found that cybervictimization doubled the tendency to depression (a*OR* 2.50, 95% *CI* [1.23, 5.08], *p* = 0.012), anxiety (a*OR* 2.38, 95% *CI* [1.29, 4.40], *p* = 0.006), and stress (a*OR* 2.85, 95% *CI* [1.41, 5.77], *p* = 0.004) whilst social media addiction was associated with a higher tendency to depression (a*OR* 1.18, 95% *CI* [1.10, 1.26], *p* < 0.001), anxiety (a*OR* 1.15, 95% *CI* [1.08,1.22], *p* < 0.001), and stress (a*OR* 1.21, 95% *CI* [1.12, 1.32], *p* < 0.001) (see Table 4).

## 4. Discussion

This study aimed to (1) determine the prevalence of cyberbullying and social media addiction; (2) determine the factors associated with cyberbullying perpetration and victimization; and (3) examine the associations between cyberbullying and social media addiction with depression, anxiety, and stress among medical students. This study is important as past studies indicated that bullying and cybervictimization are prevalent among medical students. The authors of the current study found that the prevalence of cybervictimization and cyberbullying perpetration during the past six months were 24.4% and 13.0% respectively. In the adjusted models, being male and having higher social media addiction levels were positively associated with cybervictimization, and being male was also associated with cyberbullying perpetration; social media addiction and cybervictimization were associated with a higher tendency to depression, anxiety, and stress symptoms.

The prevalence of cyberbullying in this study was within the range of other studies done among young adults in Malaysia using the same instrument, with the cyberbullying perpetrators ranging from 8 to 36% and cybervictims ranging from 18.6 to 45% [15,16,38,45]. The prevalence of cyberbullying victimization in this study is lower than in studies among tertiary students using other instruments, which found that 61% to 66% of the participants had been cyberbullied [17,46]. This may be due to cyberbullying in this study being limited to the past six months.

Regarding social media addiction, the results of this study indicated high levels of social media addiction, which is consistent with previous studies conducted on Facebook addiction, which showed that up to 47% to 61% of tertiary students were addicted to Facebook [47,48]. This may be due to the long duration of engagement with a social networking site (an average of three hours) once the user has logged in [49]. Asian students have been shown to have higher risks of social media addiction than students in the United States [50]. This may be linked to the increased availability and easy access to the internet and social media via smartphones and other electronic devices [51].

Social media addiction was positively associated with a higher tendency to cybervictimization and cyberbullying perpetration in the bivariate analyses. This may be linked to the fact that social media was found to be the primary tool for cyberbullying behavior [15,52]. Similarly, in the present study, social media appeared as the primary tool for cyberbullying with 241 respondents, or 89.3%, perceiving that cyberbullying takes place via social media, followed by 71% via messaging services, chats (45.9%), mobile phone calls (44.4%), and emails (22.2%). The relative anonymity afforded by social media platforms may have facilitated increased cyberbullying behaviors on social media as opposed to emails or personal chats [53].

Males had a higher tendency to report cybervictimization and cyberbullying perpetration. The results are consistent with several studies conducted among university students [54] and other populations such as working adults [55] and adolescents [56]. However, the research on gender in cyberbullying has not been consistent, as some studies found no difference in cyberbullying and cybervictimization among male and female students [11,57], whereas some studies found that female students were more prone to be cybervictims [58]. Nevertheless, it is concerning that male medical students had more than four times higher tendency to be a cyberbullying perpetrator and twice the tendency to cybervictimization, signifying a need to intervene with male medical students.

Regarding the motive for cyberbullying, psychological factors (attitude, entertainment, and empowerment) were found to be associated with more than twice the higher tendency to cyberbullying perpetration. The findings of our study were similar to previous literature indicating positive attitudes towards cyberbullying [59], cyberbullying for fun [38,57], and gaining power [45,60] as predictors of cyberbullying perpetration. This may be because positive attitudes towards cyberbullying were associated with lower empathy towards victims and more accepting attitudes toward cyberbullying for revenge (they deserved it) [59].

In this study, cybervictimization was associated with double the tendency to depression, anxiety, and stress symptoms. This is consistent with several studies among university students which found that cybervictims reported higher negative psychological sequelae such as depression, anxiety, and psychological distress [61]. The experience of cybervictimization may produce feelings of anger, helplessness, or hopelessness, leading to mental health problems [62]. This study also found that social media addiction was associated with a higher tendency to have depression, anxiety, and stress symptoms, consistent with another study among university students in Hong Kong [21]. That social media addiction continues to be significant after adjusting for cyberbullying shows that there may be other factors influencing the relationship between social media addiction and psychopathological symptoms, such as interrupted sleep, decreased physical activity, and fewer face-to-face social interactions [21].

The study has implications for intervening against cyberbullying and social media addiction among medical students. Considering that a quarter of medical students in this study have experienced cybervictimization and about a third may be addicted to social media, along with the negative psychological sequelae associated with these phenomena, policies, and guidelines against cyberbullying need to be enacted. Digitally informed psychiatry education should be taught, including topics on cyberbullying and social media addiction [63]. There is also a need to further examine other areas of bullying perpetration and aggression that the cyberbullying perpetrators may be involved in, considering they may become medical professionals serving vulnerable populations in the future.

This study has some limitations. First of all, the authors did not examine the role of bully-victims, but it is possible that a significant number of students were both cyberbully-cybervictims. The cross-sectional nature of this study precluded deriving causal relationships between cyberbullying, social media addiction, and depression, anxiety, and stress symptoms. Convenience sampling using an online questionnaire may lead to self-selection of the participants and bias. Our data was limited to 270 medical students from a public university in Kuching, Malaysia, and therefore there is limited generalizability of the results. Studies on this topic, conducted across all Malaysian states, are therefore necessary in the future. We did not take into account the influence of the COVID-19 pandemic, which may have facilitated technology-related problems as reliance on technology increased during this period of lockdown, and adversely affected the students’ psychological well-being. The COVID-19 pandemic may also have affected the rigor of the study, preventing us from carrying out randomized surveys. Future research should involve longitudinal studies to determine the relationships between cyberbullying, social media addiction, and psychological distress. Rigorous statistical analysis involving structural equation modelling may also be considered in future studies, and experimental studies could be carried out to test interventions which aim to alleviate cyberbullying among medical students.

## 5. Conclusions

This study found that the prevalence of cyberbullying victimization was 24.4%, whilst past-six-month cyberbullying perpetration was experienced by 13.0% of the study participants. Cybervictimization and social media addiction were associated with depression, anxiety, and stress. Males had a higher tendency to experience both cyberbullying perpetration and victimization, and individuals reporting a higher positive attitude toward cyberbullying and gaining power were associated with cyberbullying perpetration. Targeted interventions are indicated to curb cyberbullying among medical students.

## Figures and Tables

**Table 1 ijerph-20-03136-t001:** Socio-demographic characteristics among study participants (*N* = 270).

Characteristics	*M*	*SD*	*n*	Percentage (%)
Age (years old)	21.98	1.54		
**Year of study**				
Year 1			30	11.1
Year 2			70	25.9
Year 3			65	24.1
Year 4			56	20.7
Year 5			49	18.1
**Gender**				
Male			66	24.4
Female			204	75.6
**Ethnicity**				
Malay			134	49.6
Chinese			44	16.3
Indian			18	6.7
Bumiputera Sarawak			80	23.7
Bumiputera Sabah			13	3.7
**Religion**				
Muslim			152	56.3
Buddhist			28	10.4
Christian			70	25.9
Hindu			14	5.2
Others			6	2.2
**Daily Internet Usage (hours)**				
<1			2	0.7
1–5			89	33.0
>5			179	66.3
**Daily Mobile Phone Usage (hours)**				
<1			4	1.5
1–5			93	34.4
>5			173	64.1
**Daily Social Media Usage (hours)**				
<1			13	4.8
1–5			161	59.6
>5			96	35.6
**Ranking of Usage**				
Social Media			72	45.3
Email			8	5.0
Mobile phone			64	40.3
Chat applications			15	9.4
**Cybervictim**				
Yes			66	24.4
No ^a^			204	75.6
**Cyberbully**				
Yes			35	13.0
No ^a^			235	87.0
**Bystander**				
Yes			220	81.5
No ^a^			50	18.5
**BSMAS**	17.05	5.05		
Yes			102	37.8
No			168	62.2
**Depression Symptoms ^b^**	4.03	4.73		
Yes			92	34.1
No			178	65.9
**Anxiety Symptoms ^b^**	3.93	4.37		
Yes			104	38.5
No			166	61.5
**Stress Symptoms ^b^**	4.86	4.83		
Yes			57	21.1
No			213	78.9

^a^ Overall cyberbullying pattern. The figures for “no” is equivalent to the “never” category. ^b^ Those who scored mild, moderate, severe, and extremely severe were categorized as “yes”, and no refers to those whose scores were “no”. BSMAS = Bergen Social Media Addiction Scale.

**Table 2 ijerph-20-03136-t002:** Factors associated with being a cybervictim in medical students.

Factors	Crude *OR*	95% *CI*	*p*-Value	Adjusted *OR*	95% *CI*	*p*-Value
Age (years old)	1.14	0.95–1.36	0.161	1.17	0.96–1.43	0.131
**Year of Study**						
Year 1	0.69	0.23–2.07	0.511			
Year 2	0.82	0.35–1.91	0.646			
Year 3	0.76	0.32–1.81	0.535			
Year 4	1.21	0.52–2.83	0.665			
Year 5	Reference group					
**Gender**						
Male	2.67	1.50–4.90	0.001 **	2.66	1.34–5.11	0.003 **
Female	Reference group			Reference group		
**Ethnicity**						
Malay	0.73	0.18–3.00	0.664			
Chinese	0.69	0.15–3.15	0.629			
Indian	1.17	0.22–6.20	0.856			
Bumiputera Sarawak	0.71	0.16–3.11	0.654			
Bumiputera Sabah	Reference group					
**Religion**						
Muslim	0.60	0.11–3.41	0.563			
Buddhist	0.55	0.08–3.73	0.537			
Christian	0.69	0.12–4.11	0.686			
Hindu	1.11	0.15–8.37	0.919			
Others	Reference group					
**Daily Internet Usage (hours)**						
<1	0.00	0.00–0.00 ^a^	0.999	0.00	0.00–0.00 ^a^	0.999
1–5	0.63	0.30–1.20	0.141	0.72	0.36–1.45	0.362
>5	Reference group					
**Daily Mobile Phone Usage (hours)**						
<1	0.92	0.09–9.07	0.943			
1–5	0.71	0.39–1.30	0.266			
>5	Reference group					
**Daily Social Media Usage (hours)**						
<1	0.47	0.10–2.24	0.339			
1–5	0.76	0.43–1.36	0.357			
>5	Reference group					
**Cyberbullying Factors**						
Technology	2.17	1.20–3.95	0.011 *	1.61	0.83–3.15	0.162
Psychology	1.73	1.20–2.49	0.003 **	1.34	0.88–2.03	0.170
Socio-cultural	1.71	0.93–3.15	0.087	1.10	0.53–2.27	0.807
BSMAS	1.12	1.05–1.19	<0.001 **	1.09	1.02–1.16	0.013 *

* *p* < 0.05; ** *p* < 0.01. ^a^ Only 2 subjects reported using the internet for less than 1 h. *OR* = Odds Ratio; *CI* = Confidence Interval; BSMAS = Bergen Social Media Addiction Scale.

**Table 3 ijerph-20-03136-t003:** Factors associated with being a cyberbully in medical students.

Factors	Crude *OR*	95% *CI*	*p*-Value	Adjusted *OR*	95% *CI*	*p*-Value
Age (years old)	0.90	0.71–1.14	0.386			
**Year of Study**						
Year 1	1.76	0.46–6.68	0.406			
Year 2	1.64	0.53–5.06	0.389			
Year 3	0.90	0.26–3.12	0.862			
Year 4	1.47	0.45–4.82	0.528			
Year 5	Reference group					
**Gender**						
Male	4.75	2.27–9.93	<0.001 ***	4.48	1.94–10.33	<0.001 ***
Female	Reference group					
**Ethnicity**						
Malay	0.43	0.82–2.24	0.316			
Chinese	0.63	0.11–3.72	0.611			
Indian	0.50	0.06–4.23	0.525			
Bumiputera Sarawak	0.92	0.17–4.91	0.925			
Bumiputera Sabah	Reference group					
**Religion**						
Muslim	0.24	0.04–1.39	0.110	0.72	0.10–5.31	0.625
Buddhist	0.24	0.03–1.92	0.178	0.85	0.09–8.50	0.746
Christian	0.41	0.07–2.52	0.339	1.52	0.19–12.13	0.852
Hindu	0.33	0.04–3.21	0.341	0.86	0.07–11.24	0.516
Others	Reference group			Reference group		
**Daily Internet Usage (hours)**						
<1	0.00	0.00–0.00 ^a^	0.999	0.00	0.00–0.00 ^a^	0.999
1–5	0.46	0.19–1.10	0.081	0.59	0.19–1.87	0.367
>5	Reference group			Reference group		
**Daily Mobile Phone Usage (hours)**						
<1	1.89	0.19–18.82	0.589	2.36	0.13–42.04	0.560
1–5	0.53	0.23–1.23	0.139	0.96	0.28–3.31	0.946
>5	Reference group			Reference group		
**Daily Social Media Usage (hours)**						
<1	0.79	0.16–3.87	0.769	1.06	0.14–8.20	0.956
1–5	0.45	0.21–0.93	0.032 **	0.50	0.19–1.33	0.166
>5	Reference group			Reference group		
**Cyberbullying Factors**						
Technology	0.96	0.47–1.96	0.899			
Psychology	2.56	1.57–4.16	<0.001 ***	2.16	1.26–3.73	0.005 **
Socio-cultural	1.05	0.49–2.25	0.898			
**BSMAS**	1.08	1.01–1.16	0.035 **	1.04	0.95–1.13	0.430

** *p* < 0.01; *** *p* < 0.001. ^a^ Only 2 subjects reported using the internet for less than 1 h. *OR* = Odds Ratio; *CI* = Confidence Interval; BSMAS = Bergen Social Media Addiction Scale.

**Table 4 ijerph-20-03136-t004:** Association between cyberbullying, social media addiction and depression, anxiety, and stress in medical students.

Variables	Depression a*OR* (95% *CI*) ^a^	Anxiety a*OR* (95% *CI*) ^b^	Stress a*OR* (95% *CI*) ^c^
**BSMAS**	1.18 (1.10–1.26) ***	1.15 (1.08–1.22) ***	1.21 (1.12–1.32) ***
**Cybervictim**			
Yes	2.50 (1.23–5.08) *	2.38 (1.29–4.40) **	2.85 (1.41–5.77) **
No	Reference group	Reference group	Reference group
**Cyberbully**			
Yes	0.54 (0.21–1.41)	-	-
No	Reference group		

* *p* < 0.05; ** *p* < 0.01; *** *p* < 0.001. ^a^ Final model adjusted for gender, daily internet usage, daily social media usage, BSMAS, cybervictimization, and cyberbullying. ^b^ Final model adjusted for ethnicity, religion, daily social media usage, BSMAS, and cybervictimization. ^c^ Final model adjusted for year of study, daily mobile phone usage, BSMAS, and cybervictimization. a*OR* = adjusted Odds Ratio; *CI* = Confidence Interval; BSMAS = Bergen Social Media Addiction Scale.

## Data Availability

The data that support the findings of this study are available from the corresponding author upon reasonable request.

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
