# Peer review of "Cyberbullying, Social Media Addiction and Associations with Depression, Anxiety, and Stress among Medical Students in Malaysia"

_ijerph, 2023, doi:10.3390/ijerph20043136_

Round 1

Reviewer 1 Report

Thank you for providing me with the opportunity to read this paper.

This study found that the prevalence of cyberbullying victimization was 24.4%, whilst past six-month cyberbullying perpetration was experienced by 13.0% of the study participants. Cybervictimization and social media addiction were associated with depression, anxiety, and stress. Males had higher odds of experiencing both cyberbullying perpetration and victimization, and individuals reporting a higher positive attitude toward cyberbullying and gaining power were associated with cyberbullying perpetration. Targeted interventions are indicated to curb cyberbullying among medical students.

I believe that this paper has a good potential to impact the literature on the topic of cyberbullying.

There is only one suggestion I would like authors to address. The word "odds" feels unnatural, it might be replaced with "tendency" or any other sinonim that authors would like to choose.

Author Response

Reviewer 1

Thank you for providing me with the opportunity to read this paper.

This study found that the prevalence of cyberbullying victimization was 24.4%, whilst past six-month cyberbullying perpetration was experienced by 13.0% of the study participants. Cybervictimization and social media addiction were associated with depression, anxiety, and stress. Males had higher odds of experiencing both cyberbullying perpetration and victimization, and individuals reporting a higher positive attitude toward cyberbullying and gaining power were associated with cyberbullying perpetration. Targeted interventions are indicated to curb cyberbullying among medical students.

I believe that this paper has a good potential to impact the literature on the topic of cyberbullying.

There is only one suggestion I would like authors to address. The word "odds" feels unnatural, it might be replaced with "tendency" or any other sinonim that authors would like to choose.

Response: Thank you for your suggestion. We have replaced all instances of “odds” with “tendency”.

Reviewer 2 Report

Dear Authors,

It was a great pleasure to read your article. The topic is very actual and extremely important. And – most important – the study design and research realization are very satisfactory.

The title is informative and accurately reflects the manuscript. The abstract is complete and stand-alone. It adequately reflects the content of the manuscript.

The research design answers the proposed research questions. The study’s methodology and execution of the study are adequate. The important aspects of the methods are clearly described.

The results are clearly organized and presented. The analysis is adequately described. Tables and other visual materials are clear and easy to interpret.

The structure of the Discussion is very clear. The interpretations are appropriate, supported by the results, and discussed with relevant literature and within the study's limits.

My only concerns refer to the Introduction section. I think it is too short, not properly supported by the literature, and important threads are missing. Please add 1) a definition of Cyberbullying. You present a definition of social media addiction. The second main variable is not defined in the Introduction; 2) some information about the specific of medical studies in general, maybe also of medical studies in Malaysia. It can be useful for the reader to answer the question of how the specificity of medical studies makes your results unique/generalizable. How the field of studies can affect the results? When you introduce this information in the Introduction section, maybe you can also use it in the Discussion section? 3) Please expand on this part: „Problematic use of social networking sites has  been associated with psychological distress, depression, and anxiety [13,14]. Both cyberbullying and social media addiction have been associated with psychological distress among university students” It is all associated, but how? What are the results of previous studies? We have here a kind of model/configuration of  connected variables. Please write more about it. The connection seems obvious, but we still need research like yours, which problematizes the problem and show patterns of risk factors and experiences of violence, such as cyberbullying. Please expand your theoretical model of this patterns before you analyze your results.

I do not doubt that this article can be published. Please just work a little on Introduction.

Kind regards

Author Response

Reviewer 2

Comment 1: My only concerns refer to the Introduction section. I think it is too short, not properly supported by the literature, and important threads are missing. Please add 1) a definition of Cyberbullying. You present a definition of social media addiction. The second main variable is not defined in the Introduction;

Response 1: Thank you for your observation and suggestions. We have now provided a definition of cyberbullying, as follows:

(ll. 44-48) Cyberbullying has been defined as bullying which is perpetrated through the medium of technology usage in online settings, such as on social media or through a text message [5,6]. Cyberbullying has also been characterized as an intentional, repeated, and aggressive act of inflicting harm on another person through electronic means of contacting the victim, and occurs over time [7,8].

Comment 2: Some information about the specific of medical studies in general, maybe also of medical studies in Malaysia. It can be useful for the reader to answer the question of how the specificity of medical studies makes your results unique/generalizable. How the field of studies can affect the results? When you introduce this information in the Introduction section, maybe you can also use it in the Discussion section?

Response 2: We have now provided more information about specific medical studies in Malaysia and elsewhere, as follows:

(ll.101-118) There is a paucity of studies addressing both cyberbullying and social media addiction among medical students. This is important as a meta-analysis showed that internet addiction, of which social media addiction is a subset [30], was detected among 30.1% of medical students [31]. Bullying in the medical education setting has been prevalent. In a review of 68 articles, 38.2% received undue pressure to produce work, and 36.1% were directed to work below their competency level [32]. Nearly half (46.2%) of trainee doctors in a UK study reported having been cyberbullied at least once [33]. Mental health among medical students has also been low. A meta-analysis found that the pooled prevalence of anxiety among medical students was 33.8% [34], whilst the pooled prevalence of depression was 28% in another meta-analysis [35]. Suicidality is a significant issue affecting medical professionals worldwide, indicating the need for early intervention [36]. A study among Malaysian undergraduate medical students found that 23.8%, 51.6%, and 15.9% reported depression, anxiety, and stress symptoms at the beginning of the semester [37]. Therefore, we aimed to determine the prevalence of cyberbullying perpetration, cyberbullying victimization, and social media addiction among medical students and their associated factors. Furthermore, this study aimed to examine the association between cyberbullying, social media addiction, and depression, anxiety, and stress among medical students.

Comment 3: Please expand on this part: „Problematic use of social networking sites has  been associated with psychological distress, depression, and anxiety [13,14]. Both cyberbullying and social media addiction have been associated with psychological distress among university students” It is all associated, but how? What are the results of previous studies? We have here a kind of model/configuration of  connected variables. Please write more about it. The connection seems obvious, but we still need research like yours, which problematizes the problem and show patterns of risk factors and experiences of violence, such as cyberbullying. Please expand your theoretical model of this patterns before you analyze your results.

Response 3: Thank you for your suggestion. We have now added the theories which were applied to the connection between cyberbullying ad social media addiction. Furthermore, we have now added more literature to expand on past studies dealing with cyberbullying and social media addiction, as follows:

(pp.73-100) Problematic social media use, including social media addiction, may pose a risk for increased cyberbullying and cybervictimization. Craig et al. [23] applied the Problem Behavior Theory and Social Learning Theory to elucidate this association. According to the Problem Behavior Theory, risk behaviors usually occur together, and certain individuals have specific profiles which make them more vulnerable to risky behaviors. For example, individuals with social media addiction may have increased exposure to cyberbullies or cybervictims due to spending more time on these sites; thus, the addiction to social media increases the likelihood of cyberbullying to occur. Based on the Social Learning Theory, individuals who are on social media platforms for a longer time may witness more aggression, and through the effect of modeling and reinforcement, may perceive aggressors to be of higher social status, and emulate these behaviors. The sense of belongingness of conforming to group norms may also reinforce group bullying behaviors. In a study across 42 countries among adolescents aged 11 to 15 years old, problematic social media use had modest to strong associations with cyberbullying and cybervictimization [23]. Structural Equation Modelling in a study among Malaysian university students showed that social media use was significantly associated with cyber engagement, and this engagement was also associated significantly with cyber harassment, cyberstalking, and cyberbullying [24].

Both cyberbullying and social media addiction have been associated with psychological distress [25]. A study among Bosnian adolescents in a state hospital showed that those diagnosed with anxiety and depressive disorders had higher scores for cybervictimization [26]. Another study among 1,691 Malaysian adolescents revealed that both be-fore and during the COVID-19 pandemic, those with depression symptoms had a higher tendency for experiencing cyberbullying [27]. A study among college students in China showed that cyberbullying in the social media and gaming contexts was associated with higher anxiety and internet addiction symptoms [28]. Among university students in Malaysia, social media use during the COVID-19 pandemic has been linked to depression and anxiety symptoms [29] and lower self-esteem [22].

Reviewer 3 Report

In this paper, the authors aim to determine the prevalence and factors associated with cyberbullying and social media addiction. They assigned 270 medical students from a public university in Kuching, Malaysia, to carry out the experiments. The authors found some interesting things that: The prevalence of cyberbullying victimization was 24.4%, whilst 13.0% reported cyberbullying perpetration over the past six months. Male gender was positively associated with both cyberbullying perpetration and cyber victimization, whilst social media addiction was positively associated with cyber victimization. Psychological motives such as positive attitude toward cyberbullying and gaining power were associated with cyberbullying perpetration. The work is interesting but not innovative enough. The detailed review comments are as follows. 

1. The contribution of this paper is not very clear and enough. The research work focusing the Malaysian population would make the contribution smaller. Moreover, the size of the population involved in this experiment was small.

2. Sufficient comparative experiments (i.e. more datasets) are lacking to prove the correctness of conclusion. The authors are suggested to undertake more abundant experiments and analysis.

 3. There are a number of studies have been published, however, a comprehensive comparison with existing latest work is missing. It is suggested that the authors cite more recent excellent papers (i.e. in the last 3 years) and strengthen the contribution part of the paper.

For example, (a) An international systematic review of cyberbullying measurements, Computers in human behavior, 2020); (b) Cyberbullying among adolescents and children: a comprehensive review of the global situation, risk factors, and preventive measures (Frontiers in public health, 2021); (c) Improving cyberbullying detection using Twitter users' psychological features and machine learning (Computers & Security, 2020) etc.

 4. There are some typos in the current manuscript. Authors are suggested to check and polish the English writing in order to increase the readability.

Author Response

Reviewer 3

In this paper, the authors aim to determine the prevalence and factors associated with cyberbullying and social media addiction. They assigned 270 medical students from a public university in Kuching, Malaysia, to carry out the experiments. The authors found some interesting things that: The prevalence of cyberbullying victimization was 24.4%, whilst 13.0% reported cyberbullying perpetration over the past six months. Male gender was positively associated with both cyberbullying perpetration and cyber victimization, whilst social media addiction was positively associated with cyber victimization. Psychological motives such as positive attitude toward cyberbullying and gaining power were associated with cyberbullying perpetration. The work is interesting but not innovative enough. The detailed review comments are as follows. 

Comment 1. The contribution of this paper is not very clear and enough. The research work focusing the Malaysian population would make the contribution smaller. Moreover, the size of the population involved in this experiment was small.

Response 1: Thank you for pointing out how we may better angle the study’s contribution. We have now amended the contribution of the study, to focus on the problem of cyberbullying and social media addiction among medical students, as follows:

(ll.101-118) There is a paucity of studies addressing both cyberbullying and social media addiction among medical students. This is important as a meta-analysis showed that internet addiction, of which social media addiction is a subset [30], was detected among 30.1% of medical students [31]. Bullying in the medical education setting has been prevalent. In a review of 68 articles, 38.2% received undue pressure to produce work, and 36.1% were directed to work below their competency level [32]. Nearly half (46.2%) of trainee doctors in a UK study reported having been cyberbullied at least once [33]. Mental health among medical students has also been low. A meta-analysis found that the pooled prevalence of anxiety among medical students was 33.8% [34], whilst the pooled prevalence of depression was 28% in another meta-analysis [35]. Suicidality is a significant issue affecting medical professionals worldwide, indicating the need for early intervention [36]. A study among Malaysian undergraduate medical students found that 23.8%, 51.6%, and 15.9% reported depression, anxiety, and stress symptoms at the beginning of the semester [37]. Therefore, we aimed to determine the prevalence of cyberbullying perpetration, cyberbullying victimization, and social media addiction among medical students and their associated factors. Furthermore, this study aimed to examine the association between cyberbullying, social media addiction, and depression, anxiety, and stress among medical students.

Comment 2. Sufficient comparative experiments (i.e. more datasets) are lacking to prove the correctness of conclusion. The authors are suggested to undertake more abundant experiments and analysis.

Response 2: Thank you, we acknowledge the limitations of our sample, and have added the following to the limitations section:

(ll. 262-264) Our data was limited to 270 medical students from a public university in Kuching, Malaysia, and therefore there is limited generalizability of the results. Studies on this topic, conducted across all Malaysian states, are therefore necessary in the future.

Comment 3. There are a number of studies have been published, however, a comprehensive comparison with existing latest work is missing. It is suggested that the authors cite more recent excellent papers (i.e. in the last 3 years) and strengthen the contribution part of the paper.

For example, (a) An international systematic review of cyberbullying measurements, Computers in human behavior, 2020); (b) Cyberbullying among adolescents and children: a comprehensive review of the global situation, risk factors, and preventive measures (Frontiers in public health, 2021); (c) Improving cyberbullying detection using Twitter users' psychological features and machine learning (Computers & Security, 2020) etc.

Response 3: Thank you for your excellent suggestions. We have now added existing latest work on this topic into the introduction section, including the ones recommended by you, as follows:

(ll. 49-59) There is growing evidence that cyberbullying is becoming more common among children and adolescents. A systematic review by Zhu et al. [10] showed that cyberbullying perpetration among children and adolescents was between 6.0% and 46.3%, and cyberbullying victimization was between 13.99% and 57.5%. Cyberbullying is not isolated to schoolchildren and adolescents but also occurs among university students and young adults with the prevalence of cyberbullying ranging from 3% to 40% in cyberbullying perpetration and 7% to 62% in cyberbullying victimization [11]. This matter is complicated by the fact that a substantial number of cyberbullies were also cybervictims. A meta-analysis by Lozano-Blasco et al. [12] reported that there is a moderate-high correlation (r = 0.428) between being both a cybervictim and a cyberbully.

(ll.91-100) Both cyberbullying and social media addiction have been associated with psycho-logical distress [25]. A study among Bosnian adolescents in a state hospital showed that those diagnosed with anxiety and depressive disorders had higher scores for cybervictimization [26]. Another study among 1,691 Malaysian adolescents revealed that both before and during the COVID-19 pandemic, those with depression symptoms had a higher tendency for experiencing cyberbullying [27]. A study among college students in China showed that cyberbullying in the social media and gaming contexts was associated with higher anxiety and internet addiction symptoms [28]. Among university students in Malaysia, social media use during the COVID-19 pandemic has been linked to depression and anxiety symptoms [29] and lower self-esteem [22].

(ll.101-113) There is a paucity of studies addressing both cyberbullying and social media addiction among medical students. This is important as a meta-analysis showed that internet addiction, of which social media addiction is a subset [30], was detected among 30.1% of medical students [31]. Bullying in the medical education setting has been prevalent. In a review of 68 articles, 38.2% received undue pressure to produce work, and 36.1% were directed to work below their competency level [32]. Nearly half (46.2%) of trainee doctors in a UK study reported having been cyberbullied at least once [33]. Mental health among medical students has also been low. A meta-analysis found that the pooled prevalence of anxiety among medical students was 33.8% [34], whilst the pooled prevalence of depression was 28% in another meta-analysis [35]. Suicidality is a significant issue affecting medical professionals worldwide, indicating the need for early intervention [36]. A study among Malaysian undergraduate medical students found that 23.8%, 51.6%, and 15.9% reported depression, anxiety, and stress symptoms at the beginning of the semester [37].

Comment 4. There are some typos in the current manuscript. Authors are suggested to check and polish the English writing in order to increase the readability.

Response 4: Thank you, we have now thoroughly checked the manuscript for typos and language issues.

Round 2

Reviewer 3 Report

The authors addressed the main concerns from the reviews, the revised version of the manuscript appears to be good. The authors are suggested to undertake more abundant experiments and analysis.

Author Response

Reviewer comment: The authors addressed the main concerns from the reviews, the revised version of the manuscript appears to be good. The authors are suggested to undertake more abundant experiments and analysis.

Author response: Thanks for your suggestion. We have now added this to the manuscript limitations and future studies section:

(ll. 342-348) The COVID-19 pandemic may also have affected the rigor of the study, preventing us from carrying out randomized surveys. Future research should involve longitudinal studies to determine the relationships between cyberbullying, social media addiction, and psychological distress. Rigorous statistical analysis involving structural equation modelling may also be considered in future studies, and experimental studies could be carried out to test interventions which aim to alleviate cyberbullying among medical students.